# Routes to diagnosis of symptomatic cancer in sub-Saharan Africa: systematic review

Tanimola Martins [iD] , Samuel William David Merriel [iD] , William Hamilton [iD]

College of Medicine and Health, University of Exeter, Exeter, UK

**Correspondence to**
Dr Tanimola Martins;
tom207@exeter.ac.uk

## ABSTRACT

**Background** Most cancers in sub-Saharan Africa (SSA) are diagnosed at advanced stages, with limited treatment options and poor outcomes. Part of this may be linked to various events occurring in patients' journey to diagnosis. Using the model of pathways to treatment, we examined the evidence regarding the routes to cancer diagnosis in SSA.

**Design and settings** A systematic review of available literature was performed.

**Methods** The Preferred Reporting Items for Systematic Reviews and Meta-Analyses guidelines were followed. Between 30 September and 30 November 2019, seven electronic databases were searched using terms relating to SSA countries, cancer and routes to diagnosis comprising the population, exposure and outcomes, respectively. Citation lists of included studies were manually searched to identify relevant studies. Furthermore, ProQuest Dissertations & Theses Global was searched to identify appropriate grey literature on the subject.

**Results** 18 of 5083 references identified met the inclusion criteria: eight focused on breast cancer; three focused on cervical cancer; two each focused on lymphoma, Kaposi's sarcoma and childhood cancers; and one focused on colorectal cancer. With the exception of Kaposi's sarcoma, definitive diagnoses were made in tertiary healthcare centres, including teaching and regional hospitals. The majority of participants initially consulted within primary care, although a considerable proportion first used complementary medicine before seeking conventional medical help. The quality of included studies was a major concern, but their findings provided important insight into the pathways to cancer diagnosis in the region.

**Conclusion** The proportion of patients who initially use complementary medicine in their cancer journey may explain a fraction of advanced-stage diagnosis and poor survival of cancer in SSA. However, further research would be necessary to fully understand the exact role (or activities) of primary care and alternative care providers in patient cancer journeys.

## BACKGROUND

Sub-Saharan Africa (SSA) is overburdened with communicable diseases, while the incidence and mortality from non-communicable diseases such as cancer are rising across the region.[1] The increase in cancer incidence

### Strengths and limitations of this study

► This is the first systematic review of the evidence relating to the routes to diagnosis of cancer in Sub-Saharan Africa (SSA).
► The search strategies, assessment of quality and narrative synthesis followed good practice.
► Selected studies used small sample sizes and systematically introduced biases in the selection of participants and data collection.
► However, their findings provide unique insights into patients' journey to cancer diagnosis in SSA.

is associated with poor control of cancer-related infections and unhealthy lifestyle choices, which may be addressed, in part, by implementing effective public health interventions.[2–6] Mortality from cancer is strongly associated with stage at diagnosis; early-stage cancers enable treatment with curative intent and better prognoses than late-stage diseases.[7–9] Most cancers in SSA are diagnosed at advanced stages due to late presentation of symptoms, weak referral mechanisms and limited diagnostic capacity.[8–10] Early-stage cancers and precancerous lesions are detectable by screening asymptomatic patients, but this is limited to few sites and is very rarely used in SSA. Therefore, interventions aimed at promoting early symptomatic presentation and expedited diagnosis are likely to yield better cancer outcomes in the region. However, such interventions must be rooted in empirical evidence to ensure effectiveness and to maximise local resources use.

The model of pathways to treatment offers a useful framework to examine the routes to diagnosis of symptomatic cancer.[11] It describes five possible events in the pathways to treatment: detection of bodily changes, perceived reasons to seek medical help, first consultation with a healthcare provider, diagnosis and start of treatment.[11] Numerous studies have explored these events in cancer, but only a few have specifically investigated patients'

initial contact with healthcare providers in SSA.[12] Using this framework, we investigated patients' routes to cancer diagnosis in SSA, focusing on the initial point of consultation and eventual diagnosis. Identifying and categorising the routes to diagnosis may explain advanced-stage cancers and provide the basis for early diagnosis interventions in the region.

## METHODS

A systematic narrative review was performed. The conduct and reporting of the review was based on the Preferred Reporting Items for Systematic Reviews and Meta-Analyses (PRISMA) framework (see online supplemental file 1).[13]

### Search strategy

Between 30 September and 30 November 2019, a systematic search of the following electronic databases was performed: Ovid MEDLINE(R) ALL (1946–30 September 2019), Embase (1974–30 September 2019), Web of Science (1915 (1)–2019 (69)), PsycINFO (1806– week 2 of September 2019), CINAHL Complete, Global Health (1973–week 36 2019) and *African Journals Online*. The search strategy included terms, their synonyms and Medical Subject Headings terms relating to SSA countries, cancer and routes to diagnosis; comprising the population, exposure and outcomes, respectively (table 1). Online supplemental file 2 shows the search strategy in MEDLINE, PsycINFO, Embase and Global Health. Citation lists of included studies were manually searched to identify relevant studies. Furthermore, ProQuest Dissertations & Theses Global was searched to identify appropriate grey literature on the subject.

### Eligibility criteria

Included studies investigated cancer diagnosis, described the routes or the patient's pathway to diagnosis (including the settings of initial consultation and definitive diagnosis) and were conducted in 1 or more of the 48 SSA countries. The list of SSA countries matches those featured on the World Bank data catalogue used to describe health and socioeconomic indices in the region.[14] Excluded studies were non-English studies, focused on populations outside the region of SSA, investigated diseases other than cancer, cancer treatment, outcomes and attitudes toward cancer diagnoses. All study designs (qualitative and quantitative) were eligible for inclusion.

### Study selection

This involved a two-stage screening process. First, title, abstract and full articles of potentially eligible studies were sequentially screened by an experienced researcher (TM) against the inclusion and exclusion criteria. Consequently, studies that appeared to meet the inclusion criteria or where a decision could not be made based on the title and/or abstract were selected for full-text review to identify those for the final analysis.

### Data extraction and synthesis

One reviewer (TM) extracted data from all included studies. Extracted data were added to a data extraction spreadsheet, which was initially piloted with seven studies. Data extraction included study characteristics: country of study, design, participants' characteristics, cancer type, healthcare settings for initial consultation and eventual diagnosis. Quantitative synthesis was not possible because our final selection differed in terms of cancer sites and

| Table 1 | Search terms | |
|---|---|---|
| **Population** | **Exposure** | **Outcome** |
| Terms relating sub-Saharan Countries Angola, Gabon, Nigeria, Benin, Gambia, The Rwanda, Botswana, Ghana, São Tomé and Principe, Burkina Faso, Guinea, Senegal, Burundi, Guinea-Bissau, Seychelles, Cabo Verde, Kenya, Sierra Leone, Cameroon, Lesotho, Somalia, Central African Republic, Liberia, South Africa, Chad Madagascar, South Sudan, Comoros, Malawi, Sudan, Congo, Dem. Rep., Mali, Swaziland, Congo Rep., Mauritania, Tanzania, Côte d'Ivoire, Mauritius, Togo, Equatorial Guinea, Mozambique, Uganda, Eritrea, Namibia, Zambia, Ethiopia, Niger, Zimbabwe | Terms relating to cancer Cancer, Neoplasm, Malignant Neoplasm, tumour, Malignant tumour, Astrocytoma, Adenocarcinoma, Glioma, Mesothelioma, Medulloblastoma, Myeloma, Melanoma, Neuroblastoma, Sarcoma, Nonmelanoma, Osteosarcoma, Teratoma, Seminoma, Hodgkin, Leukaemia, Lymphoma, Retinoblastoma | Terms relating to the routes to cancer diagnosis: Pathway to diagnosis Pathway to detect* Routes to diagnos* Routes to detect* Diagnos* Detect* Consult* Help-seek* Present* Route to consult* Routes to present* Pathway to consult* Pathway to present* Primary care Family doctor Physician Healthcare practitioner General Practitioners Family Practice Primary Healthcare |

outcome measures. For instance, some studies described patients initially presenting to 'healthcare practitioner', a term that may be used to describe primary care physicians or doctors in secondary care. Therefore, we performed a narrative synthesis using the framework of Rodgers and colleagues.[15] Participants' characteristics and the study's main findings are illustrated in tables and figures.

## Quality assessment

Three reviewers (TM, WH and SWDM) assessed the methodological quality of eligible studies using the Newcastle-Ottawa Quality Assessment Scale (NOS) for cohort, NOS adapted for cross-sectional studies,[16 17] and the Joanna Briggs Institute (JBI) Critical Appraisal Checklist for Qualitative Research.[18] TM and SWDM independently selected the appropriate checklist based on study design. The cohort and cross-sectional studies were awarded stars and rated 'good', 'satisfactory' or 'poor quality', depending on the extent to which they meet the NOS checklist criteria on the three main domains: selection, comparability and outcomes alongside associated statistics. Good-quality studies were awarded four stars in the selection domain, and two stars in each of the comparability and outcome domains. Studies rated satisfactory were awarded two stars in the selection domain, one star in comparability domain, and up to three stars in the outcome domain. Poor-quality studies were awarded zero star in the comparability domain, and one star in the selection or outcome domains. The JBI checklist is not a scoring system but a useful tool for evaluating the risk of bias in the design and conduct of qualitative studies. The checklist consists of 10 criteria with four possible responses: 'yes,' 'no,' 'unclear' and 'inapplicable.' Each qualitative study was evaluated

against the checklist criteria. Discrepancies between the reviewers were resolved by consensus, although no study was excluded based on quality.

## Patient and public involvement

There was no formal patient and public involvement in this review.

## RESULT
### Study characteristics

The search identified 5083 articles. After screening title and abstract and removing duplicates, 4933 irrelevant articles were excluded: 150 full-text articles were assessed with 18 meeting the inclusion criteria. A PRISMA flowchart showing the reasons for abstract and full article exclusions is shown in figure 1. The 18 studies recruited a total of 4871 participants from nine SSA countries, 70% of which were females with the average age ranging from 4 to 59 years. The characteristics of included studies are illustrated in table 2, with the results of quality assessment in table 3. Seven of the studies were conducted in Nigeria,[19–25] three in Ethiopia,[26–28] two each in Ghana[29 30] and South Africa,[31 32] and one each in Cameroon,[33] Tanzania[34] and Kenya.[35] The final study involved five countries (Kenya, Uganda, Malawi, Cameroon and Nigeria).[36]

All 18 studies were observational with 7 cross-sectional surveys, 7 cohorts (using medical records), 3 qualitative (face-to-face interviews) and a mixed-methods study (using both qualitative and quantitative data). Eight studies examined breast cancer[19–22 26 27 29 30]; three focused on cervical cancer[23 28 34]; two each focused on lymphoma,[32 33] Kaposi's sarcoma[31 36] and childhood cancers[25 35]; and one

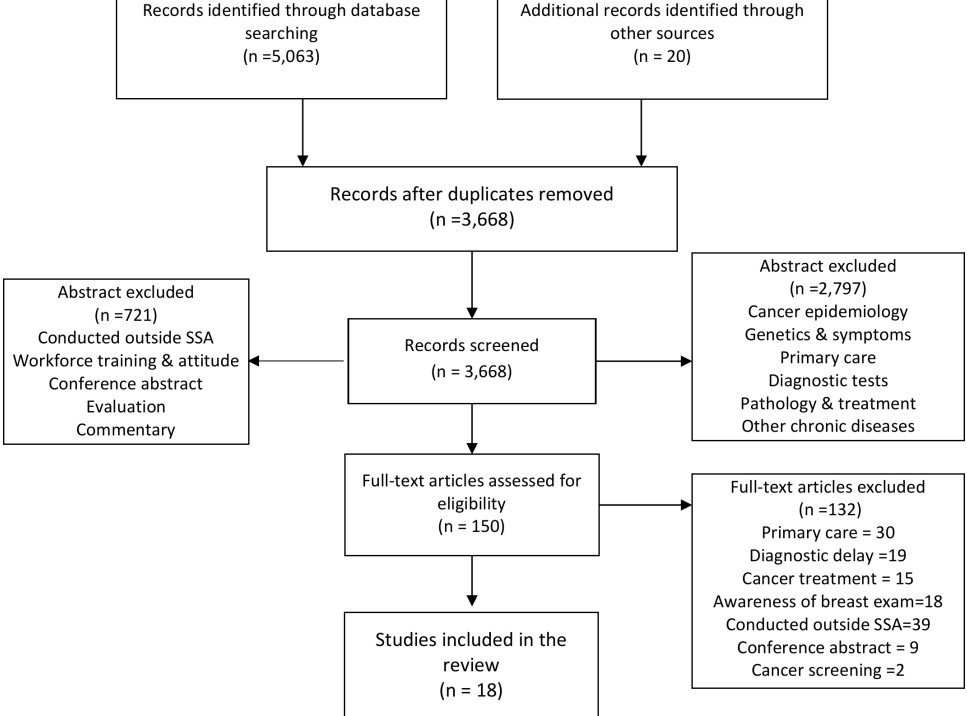

**Figure 1** Flowchart of the study selection process. SSA, sub-Saharan Africa.

**Table 2** Study characteristics

| Author | Country | Type/site | Title | Method | Outcome measure | Sample characteristics | Relevant findings |
|--------|---------|-----------|-------|--------|-----------------|------------------------|-------------------|
| Dye et al[26] | Ethiopia | Breast | Complex care systems in developing countries: breast cancer patient navigation in Ethiopia | A mixed-methods study using semistructure interview to investigate the participant's navigation through the health system before arriving at the tertiary health centre for treatment | Patient navigation through the healthcare system that culminated in the treatment of cancer | Participants: 55 patients with breast cancer plus 14 carers<br>Mean age: 45.5 years<br>Sex: 98% female | Initial presentation of symptoms was 53.7% to primary care, 16.4% to traditional healers, 16.4% to local/regional hospital, 9% to private hospital and 4.5% directly to the tertiary referral centre<br>Definitive diagnoses were made at the tertiary health centre. |
| Jemebere[27] | Ethiopia | Breast | Barriers associated with presentation delay among breast cancer patients at Hawassa University Comprehensive and Specialised Hospital, Southern Ethiopia | Cross-sectional survey of women diagnosed with breast cancer at a specialised hospital | Route to diagnosis | Participants: 106 women<br>Age range: 15–65 years<br>Occupation: farmers (3%), labourer (9%), merchant (16%), professional 29%) and housewife (43%)<br>Education: none (28%), elementary (29%), high school (24%) and college (≥19%)<br>Family history of breast cancer (13%) | 64% delayed presenting to the hospital due to initial use of complementary medicine, including herbal remedy, traditional healers and prayers<br>Definitive diagnoses were made at the teaching hospital. |
| Ezeome[19] | Nigeria | Breast | Delays in presentation and treatment of breast cancer in Enugu, Nigeria | Cross-sectional survey of patiets with breast cancer at an oncology specialist unit (in a teaching hospital) | Patients first point of symptom(s) presentation<br>Patients first point of conventional medical treatment | Participants: 164 patients (162 female and 2 male)<br>Median (range) age: 45 (21–77) years<br>Socioeconomic status: low (59%), middle (40%), high (1%)<br>Education: none (15%), primary (24%), secondary (29%), degree (30%)<br>Religion: 96% Christians | 13.1% first used traditional healers.<br>4.4% first presented to a prayer house.<br>82.3% initially presented to healthcare facilities.<br>First point of contact with medical facilities: 50% within primary (GP), 25% to consultant surgeon, 10.1% to patent medicine dealer, 7.5% to a gynaecologist, 5% to a nurse or allied health professional |

Continued

**Table 2** Continued

| Author | Country | Type/site | Title | Method | Outcome measure | Sample characteristics | Relevant findings |
|---|---|---|---|---|---|---|---|
| Pruitt et al[20] | Nigeria | Breast | Social barriers to diagnosis and treatment of breast cancer in patients presenting at a teaching hospital in Ibadan, Nigeria | A qualitative study which used semistructured interview of patients with breast cancer in a teaching hospital | Help-seeking behaviour after noticing symptoms. | Participants: 31 women with breast cancer Median (range) age: 51 (28–≥80) years Education: none (n=7), primary/secondary (n=15), tertiary (n=9) Religion: Christians (83%) and Muslim (17%) | Most women rapidly sought orthodox medical care once they noticed symptoms, but few reported seeking herbal/spiritual help initially. Definitive diagnoses were made at the teaching hospital. |
| Adesunkanmi et al[21] | Nigeria | Breast | The severity, outcome and challenges of breast cancer in Nigeria | A retrospective study using 8 years records of patient with breast cancer diagnosis in a tertiary health centre. | To determine the challenges of breast cancer diagnosis at the centre | Participants: 212 patients Sex: 99.5% female Mean (SD) age: 48±12.3 years. Occupation: traders (52%), teachers (31.6%), nurses (5.5%), farmers (4%) and self-employed (1.4%) Education: primary (18%), secondary (14%) and tertiary (35%) Previous breast disease (25%), family history of breast cancer (7.2%) | ▲ 92% of the tumour was self-detected. ▲ 4.2% was detected by physicians. ▲ 3.8% was detected by partners. ▲ Definitive diagnoses of all cases were made at the tertiary health centre. |
| Ahmed et al[22] | Nigeria | Breast | Management and outcomes of male breast cancer in Zaria, Nigeria | A retrospective study using 10 years of medical records of men with breast cancer in a specialist oncology centre (in a teaching hospital) | Route to diagnosis | Participants: 57 men Mean (SD) age: 59±2.3 years | ▲ Definitive diagnoses were made at the specialist centre. ▲ 21% initially consulted traditional healers before presenting to the specialist. ▲ 49% first presented symptoms to the specialist. |

Continued

**Table 2** Continued

| Author | Country | Type/site | Title | Method | Outcome measure | Sample characteristics | Relevant findings |
|---|---|---|---|---|---|---|---|
| Aziato and Clegg-Lamptey[29] | Ghana | Breast | Breast cancer diagnosis and factors influencing treatment decisions in Ghana | A qualitative study using face-to-face interviews of patients with breast cancer patients from a surgical unit and breast cancer support group | Route to diagnosis | Participants: 12 women Age range: 31–60 years Religion: all were Christians. | Women self-identified breast lesion or accidentally during medical examination for other problems. Participants with self-identified breast lesion presented directly to the tertiary health centre where definitive diagnoses were made. |
| Agbokey et al[30] | Ghana | Breast | Knowledge and health seeking behaviour of breast cancer patients in Ghana | A qualitative study using in-depth interviews to examine help-seeking behaviour of female breast cancer patients at a teaching hospital | Patient help-seeking behaviour after noticing female breast cancer symptoms | Participants: 20 women Median (range) age: 52.5 (29–80) years Occupation: traders (n=11), teachers (n=3), farmers (n=5) and nurse (n=1). Education: none (n=4), primary (n=12), secondary (n=1) and tertiary (n=3) Religion: 95% were Christians. | 12/20 first sought unorthodox care (herbalist, drug stores, home remedies and prayer camps) after noticing symptoms. Some patients went through a cycle of hospital-to-herbalist and back to hospital care before diagnosis. In all of the cases, however, definitive diagnoses were made at the tertiary health centre. |
| Mlange et al[34] | Tanzania | Cervical | Patient and disease characteristics associated with late tumour stage at presentation of cervical cancer in north-western Tanzania | A cross-sectional survey of women with histologically confirmed cervical cancer at a tertiary health centre | Route to diagnosis | Participants: 202 women with cervical cancers Mean (SD) age: 50±11 years Education: none (57%), primary (39%), secondary (2.4%) and college (1.4%). Occupation: farmers (84%), trader (9.9%), employed (2.4%), business (0.9%) and unemployed (2.4%) | ▲ 95 (47%) initially presented to a traditional health practitioner. ▲ Presenting to a traditional health practitioner was strongly associated with late-stage diagnosis (OR=2.3, 95% CI 1.2 to 4.2, p=0.011) ▲ Definitive diagnosis was made at the tertiary health centre. |
| Begoihn et al[28] | Ethiopia | Cervical | Cervical cancer in Ethiopia: predictors of advanced stage and prolonged time to diagnosis | A retrospective cohort study of patients diagnosed with primary cervical cancer at a tertiary health centre | Route to diagnosis of cervical cancer | Participants: 1575 women with cervical cancers Mean (SD) age: 48.9±11.5 years | All of the 1575 women presented with cervical cancer symptoms to the tertiary health centre. |

Continued

**Table 2** Continued

| Author | Country | Type/site | Title | Method | Outcome measure | Sample characteristics | Relevant findings |
|---|---|---|---|---|---|---|---|
| Eze et al[23] | Nigeria | Cervical | A six-year study of the clinical presentation of cervical cancer and the management challenges encountered at a state teaching hospital in Southeast Nigeria | A retrospective cohort study of patients diagnosed with primary cervical cancer at a tertiary health centre | Route to diagnosis of cervical | Participants: 61 women with primary cervical cancers. Mean (SD) age: 54±12.7 years Education: none (36.1%), primary (39.3%), secondary (23%) and college (1.6%) Occupation: farmers (60%), trader (37.7%), dependent (36.1%), business (0.9%) and retired (6.6%) | All of the 61 women presented with cervical cancer symptoms to the tertiary health centre. |
| Alatise et al[24] | Nigeria | Colorectal | Health-seeking behaviour and barriers to care in patients with rectal bleeding in Nigeria | A prospective survey of patients with rectal bleeding in the general population. | Attitude about seeking expert opinion among patients with rectal bleeding Initial help-seeking after the onset of rectal bleeding | Participants: 82 patients with rectal bleeding Median (range) age: 45 (18–85) years Sex: 78% were men Education: primary (28%), secondary (33%), tertiary (30%) Religion: Christians (66%) and Muslims (33%) | ▲ 39% of the participants consulted a physician with rectal bleeding. ▲ 38% suggested that herbs should be used before seeing a physician. ▲ Patients who scored high on knowledge of rectal bleeding were more likely to consult the physician (OR: 3.82; 95% CI, 55 to 10.2). |
| Chu et al[31] | South Africa | Kaposi's sarcoma | AIDS-associated Kaposi's sarcoma is linked to advanced disease and high mortality in a primary care HIV programme in South Africa | Analysis of data from a cohort study of patients with AIDS-associated Kaposi's sarcoma in primary care | Patient pathway to diagnosis of Kaposi's sarcoma | Participants: 215 patients with Kaposi's sarcoma Median age: 34 (IQR 29–41) years Sex: 41% women | 189/6292 patients enrolled at the HIV clinic were diagnosed with AIDS-associated Kaposi's sarcoma during routine examination. |
| Freeman et al[36] | Kenya Uganda Malawi Nigeria Cameroon | Kaposi's sarcoma | Pitfalls of practicing cancer epidemiology in resource-limited settings: the case of survival and loss to follow-up after a diagnosis of Kaposi's sarcoma in five countries across sub-Saharan Africa | Analysis of HIV-infected patients' in primary care records across five countries | Route to diagnosis of Kaposi's sarcoma | Participants: 1328 patients with Kaposi's sarcoma Median age: 35 (IQR 30–41) years Sex: 40% women | During routine examination for AIDS-related infections at the HIV clinic, 1328 patients were diagnosed with Kaposi's sarcoma across the five countries between 2009 and 2012. |

Continued

**Table 2** Continued

| Author | Country | Type/site | Title | Outcome measure | Method | Sample characteristics | Relevant findings |
|---|---|---|---|---|---|---|---|
| Afungchwi et al[33] | Cameroon | Burkitt lymphoma | The role of traditional healers in the diagnosis and management of Burkitt lymphoma in Cameroon: understanding the challenges and moving forward | Route to diagnosis Burkitt lymphoma | A survey of parents and carers of children diagnosed with Burtkitt lymphoma in three large hospitals | Participants: 384 completed the questionnaire. Median age: 8 (range 1–15) years Sex: male (57.4%) and female (42.4%) Religion: Christians (68.9%) and Muslims (30%) | ▲ Overall, 55% of parents used traditional healers before hospital admission. ▲ 41.8% first consulted traditional healers before reporting at the local health centre. |
| Antel et al[32] | South Africa | Lymphoma | The determinants and impact of diagnostic delay in lymphoma in a TB and HIV endemic setting | Route to diagnosis of Hodgkin and non-Hodgkin's lymphoma | A retrospective cohort study of patients diagnosed with lymphomas Data sources included hospital records, telephone and face-to-face interviews. | Participants: 163 HIV patients Median age: 48 (range 15–86) years Sex: 58% male Socioeconomic status: 70% on social grant or <251 monthly income | All 163 HIV patients were diagnosed with Hodgkin (41) and non-Hodgkin (122) lymphoma. They were referred to the tertiary health centre by healthcare practitioners. |
| Brown et al[25] | Nigeria | Childhood | A Prospective Study on the Causes of Delayed Diagnosis of Childhood Cancer in Ibadan, Nigeria | Factors influencing pre-diagnostic intervals among parent/carers of patient with childhood cancers | A survey of parents and carers of children diagnosed with malignant tumour in a tertiary healthcare settings | Participants: 91 children with cancer Median (range) age: 4 years (1 month –15 years). Sex: 50.5% male | ▲ 69% of parents initially sought medical help for their children within a health facility ▲ 19% self-medicated. ▲ 4% used an herbalist. ▲ 3% consulted a patent medicine dealer. ▲ 2% presented to a nurse/health worker and 1% visited a church. Health facilities used comprised 69% public hospital, 31% private |

Continued

**Table 2** Continued

| Author | Country | Type/site | Title | Method | Outcome measure | Sample characteristics | Relevant findings |
|---|---|---|---|---|---|---|---|
| Njuguna *et al*[35] | Kenya | Childhood | Factors influencing time to diagnosis and treatment among paediatric oncology patients in Kenya | A cross-sectional survey of parent and carers of 99 children diagnosed with a malignancy. | Help-seeking after the onset of symptoms. | Participants: 99 children with cancer Median age: children: 5.7 years, mother: 31 (19–56) years Sex: 67% male Religion: 99% of mothers were Christians. Employment: farmers (29%), regular jobs (24%), casual labourers (6%) and unemployed (6%) | 58 (59%) of parents initially sought alternative treatment for their children, including praying ceremonies (41%), visiting herbalist (36%), special food intake (11%) and attending traditional healer (3%) First contact with conventional healthcare facilities included 60% in primary care, 38% in secondary and 2% in tertiary healthcare. |

GP, general practitioner.

focused on colorectal cancer.[24] None of the 18 studies specifically investigated the routes to cancer diagnosis, although 15 studies reported the settings of initial consultation after symptom onset. The remaining three studies recruited participants from primary care-based HIV clinics to investigate Kaposi's sarcoma and lymphoma diagnoses.[31 32 36] These studies were included in our final selection, given that both cancer types are significantly more common in patients with HIV and that patients with HIV are mostly seen at such settings.

### Assessment of study quality

Overall, none of the qualitative studies fulfilled the JBI checklist criteria, and none of the quantitative studies could be classified as 'good quality' due to the limitations in their methodology (table 3). The main limitations of these studies pertained to their small sample sizes, biases in participant recruitment and data collection strategies. The sample sizes in most of the cohort and cross-sectional studies were rather small to be representative of the target population. Four-fifth of included studies recruited participants from tertiary healthcare centres, thereby introducing selection bias by systematically excluding patients diagnosed or treated elsewhere. In some studies, surveys and face-to-face interviews were performed by nurses or physician–researchers from the hospitals where participants were undergoing treatment, thus drawing possibly desirable responses. Additionally, statistical analyses were largely descriptive, with most studies presenting percentages only. Despite these limits, however, the studies provided some important findings relevant to the aim of our review, thereby warranting their inclusion in the synthesis.

### Routes to cancer diagnosis

Across the eight studies on breast cancer, providers in tertiary healthcare centres made the definitive diagnoses in all cases (table 2).[19–22 26 27 29 30] After noticing symptoms, participants initially consulted the physicians (in primary or secondary care), used complementary medicine (including traditional healers, herbalists and prayer centres) or presented directly to the hospital. The proportion of patients using each of these routes to diagnosis differed slightly between studies but very similar across all the eight studies.[19–22 26 27 29 30] On average, around a third of the participants—across the studies—initially presented with symptoms to each of the physician, complementary medicine practitioners or directly to the hospital.

In two of the three studies focused on cervical cancer, participants presented with symptoms directly to tertiary health centres where cervical cancer diagnoses were confirmed (table 2).[23 28] Conversely, 47% of the participants in the third study initially presented symptoms to traditional healthcare practitioners before returning to the tertiary health centres for diagnosis and start of treatment.[34]

**Table 3** Quality of studies

**Study quality and score based on NOS for cohort studies**

| Author, cohort studies | Selection | | | | Comparability | Outcomes and associated statistical analysis | | | Overall quality |
|---|---|---|---|---|---|---|---|---|---|
| | Representativeness of the exposed cohort (★) | Selection of the non-exposed cohort (★) | Ascertainment of exposure (★) | Demonstration that outcome of interest was not present at start of study (★) | Comparability of cohorts on the basis of the design or analysis (★★) | Assessment of outcome (★) | Was follow-up long enough for outcomes to occur? (★) | Adequacy of follow-up of cohorts (★) | |
| Ahmed et al[22] | ★ | – | – | ★ | ★ | ★ | ★ | – | Sat |
| Adesunkanmi et al[21] | ★ | – | ★ | ★ | – | ★ | ★ | – | Poor |
| Begoihn et al[28] | ★ | – | ★ | -★ | ★ | ★ | -★ | – | Sat |
| Chu et al[31] | ★ | – | ★ | ★ | ★ | ★ | ★ | – | Sat |
| Freeman et al[36] | ★ | – | ★ | ★ | ★ | ★ | ★ | – | Sat |
| Eze et al[23] | ★ | – | ★ | ★ | – | ★ | – | – | Poor |
| Antel et al[32] | ★ | – | ★ | ★ | ★ | ★ | ★ | – | Sat |

**Study quality and score based on NOS adapted for cross-sectional studies**

| Author, cross-sectional studies | Representativeness of the sample (★) | Justification of sample size (★) | Non-respondents (★) | Ascertainment of exposure (★★) | Comparability of subjects (★★) | Assessment of outcome (★★) | Statistical test (★) | Overall quality |
|---|---|---|---|---|---|---|---|---|
| Jemebere[27] | ★ | – | ★ | ★ | – | ★ | – | Poor |
| Ezeome[19] | ★ | – | – | ★ | – | ★ | – | Poor |
| Mlange et al[34] | ★ | – | ★ | ★ | ★ | ★ | ★ | Sat |
| Alatise et al[24] | ★ | – | – | ★ | – | ★ | – | Poor |
| Afungchwi et al[33] | ★ | – | – | ★ | – | ★ | – | Poor |
| Brown et al[25] | ★ | – | – | ★ | ★ | – | – | Poor |
| Njuguna et al[35] | ★ | – | ★ | ★ | ★ | ★ | – | Sat |

**Study quality and scores based on the JBI Checklist for Qualitative Research**

| Authors, qualitative studies | Congruity between stated philosophical perspective and the research methodology? | Congruity between research methodology and the research question or objectives? | Congruity between research method and the methods used to collect data? | Congruity between research method and representation and analysis of data? | Congruity between method and interpretation of results? | Statement locating the researcher culturally or theoretically? | Influence of the researcher on the research, and vice versa, addressed? | Participants' voices adequately represented? | Evidence of ethical approval? |
|---|---|---|---|---|---|---|---|---|---|
| Dye et al[26] | Unclear | Yes | Yes | Yes | Yes | Unclear | No | Yes | Yes |
| Pruitt et al[20] | Unclear | Yes | Yes | Yes | Yes | Unclear | No | Yes | Yes |

Continued

**Table 3** Continued

Study quality and scores based on the JBI Checklist for Qualitative Research

| Authors, qualitative studies | Congruity between stated philosophical perspective and the research methodology? | Congruity between research method and the research question or objectives? | Congruity between research method and the methods used to collect data? | Congruity between research method and the representation and analysis of data? | Congruity between method and interpretation of results? | Statement locating the researcher culturally or theoretically? | Influence of the researcher on the research, and vice versa, addressed? | Participants' voices adequately represented? | Evidence of ethical approval? |
|---|---|---|---|---|---|---|---|---|---|
| Aziato and Clegg-Lamptey[29] | Unclear | Yes | Yes | Yes | Yes | Unclear | No | Yes | Yes |
| Agbokey et al[30] | Unclear | Yes | Unclear | Yes | Yes | Unclear | No | Yes | Yes |

JBI, Joanna Briggs Institute; NOS, Newcastle–Ottawa Quality Assessment Scale; Sat, satisfactory-quality paper.

In a survey of 82 patients with rectal bleeding and colorectal cancer, Alatise *et al* found that only 39% of the participants had consulted a physician, with 38% of participants opting to use herbs before going to the doctors (table 2).[24]

Of 6292 HIV-infected patients enrolled at an HIV clinic, Chu *et al*[31] found 3% diagnosed with Kaposi's sarcoma within 7 years of routine HIV care. Similarly, healthcare providers from 33 HIV clinics across five African countries diagnosed 1328 HIV patients with Kaposi's sarcoma during 4 years of routine HIV care.[36] In both studies, providers at the HIV clinics detected Kaposi's sarcomas during routine examination for opportunistic infections.

Two studies surveyed parents and carers of children with childhood cancers to determine causes of diagnostic delay. In one study, 59% of parents initially sought complementary medicine for their children, although about 60% later consulted in primary care, 38% in secondary care and 2% presented directly to tertiary care.[35] In contrast, 69% of parents in the second study initially sought conventional medical help, but 24% either self-medicated, used herbalist services or presented to a church.[25]

In a survey of parents and carers of children with Burkitt lymphoma, Afungchwi and colleagues showed that 55% had used traditional healers before hospital admission, with 42% using this service before reporting to primary care.[33] In contrast, all 163 patients diagnosed with Hodgkin and non-Hodgkin's lymphoma in Antel *et al*'s[32] study were referred to the specialist by healthcare practitioners.[32]

## DISCUSSION

The route to diagnosis is a strong predictor of cancer outcomes.[37 38] In this review, we examined the evidence relating to cancer diagnosis in SSA. Across all selected studies, definitive diagnoses of cancer were made by specialists in large tertiary healthcare centres, except for Kaposi's sarcomas, which were diagnosed at various primary care-based specialist clinics. However, participants' journeys to the specialist clinics are often indirect, with a considerable proportion initially using complementary medicine before consulting conventional medical services.

### Strengths and limitations

To our knowledge, this is the first systematic review of the evidence regarding the routes to cancer diagnosis in SSA. Our rigorous search strategy and explicit inclusion/exclusion criteria, quality assessment of included studies, and narrative synthesis followed good practice. Our search identified only a modest number of studies, a third of which were conducted in Nigeria, the most populous country with the largest economy in the region. We omitted non-English studies as these may include studies published in French, Portuguese and other African languages. While the decision to omit these studies may have reduced the number of selected studies slightly, we

have no reason to believe that such omission had any impact on our findings.

About half of our final selection focused on breast cancer, reducing the scope of the review. The studies also had small sample sizes, which limits the interpretation and generalisability of our findings. Additionally, the majority recruited participants and gathered data (using researcher-administered questionnaires) from the hospital facilities where patients were being treated for their cancers, typically in the tertiary healthcare centres. This is not surprising, given the weak primary care and limited cancer registries in SSA, thus limiting the quality and quantity of data available for research. However, recruiting participants from tertiary health centres systematically exclude patients treated in private hospitals and those whose cancers may never be found due to affordability or comorbidity. Furthermore, gathering data from the hospital using physician-administered questionnaires may generate more socially desirable responses. In this case, it is likely that participants under-report their use of complementary medicine and self-medication to look good in the eyes of their providers, who may be part of the research team.

Finally, publication bias is possible as some studies on the subject may have failed to be published in reputable peer-reviewed journals, and so would have been omitted from the databases searched for this review.

### Interpretation of findings

The pathways to diagnosis of symptomatic cancer involves a series of events, beginning with the patient noticing a bodily change and deciding to seek medical help.[11] Definitive diagnosis requires biopsy of affected tissue by specialists in secondary or tertiary healthcare settings. In high-income countries like the UK and Denmark, most patients with cancer initially present with symptoms to primary care, with a smaller proportion presenting to secondary care as emergencies.[37] Primary care physicians in these countries play a key role in selecting those whose symptoms warrant specialist investigations using preliminary test results and clinical guidelines.[39] Healthcare services in many SSA countries are pluralistic, comprising a three-tier system: primary care (including dispensaries, health centres and private clinics); secondary care (including private, mission and district hospitals); and tertiary healthcare.[26] The tertiary healthcare centres are referral centres with various subspecialties and are the main setting for definitive diagnosis of cancer.[26 28 30 40] However, the role of primary care in SSA is not always well defined, with several unorthodox providers, including traditional healers and faith clinics, offering similar services, although unqualified to diagnose cancer or to refer patients for specialist investigations.[9 41–43] Patients in these countries may present with symptoms directly to tertiary healthcare centres, regardless of the nature or duration of symptoms. They may also be referred by physicians in primary or secondary care, but often with no standardised referral pathways or mechanism to

ensure continuity of care.[9 41–43] This problem is further compounded by frequent long distances to healthcare centres and out-of-pocket payments, particularly for patients in rural and socioeconomically deprived areas who may resort to complementary medicine instead.

Indeed, a considerable proportion of participants in this review initially used complementary medicine before consulting in primary care, with some also presenting directly to the hospital. Only a third of women with breast cancer initially reported symptoms to primary care, despite widespread awareness campaign with relatively easy to spot symptoms.[44 45] Fifty-three per cent of patients with cervical cancer symptoms, 39% of those with rectal bleeding and around two-thirds of childhood cancers initially sought help in primary care. Access to conventional healthcare is restricted in most SSA countries due to limited availability and affordability.[9 42 46 47] In their respective cancer journey, patients in this region may start with or revert to complementary medicine, which is considered cheaper and more natural, with some practitioners offering complete cure of cancer rather than possible remission offered by conventional medicine.[26 41] The use of complementary medicine is widespread in SSA, although evidence suggests that the practitioners can misdiagnose cancer, resulting in advanced-stage diagnosis and reduced chances of survival.[33 48]

The findings of this review may have been influenced by the level of bias in included studies: in which case, our report on the proportion using various routes to diagnosis will be inaccurate. If at all, we may have overestimated the proportion of patients consulting in primary care or underestimated those using complementary medicine before diagnosis, given the lack of public awareness of cancer and weakness of healthcare systems in the region, with significant underdiagnoses.

### CONCLUSION

Recent data from SSA suggest a rapid increase in the risk and deaths from major cancer types. In a region where infectious diseases persist, with limited healthcare budgets and shortages of specialists, urgent solutions are required to minimise the burden of cancer on its rapidly growing and ageing population. The majority of participants in our selected studies initially presented symptoms to primary care, though the proportion first using complementary medicine is considerable. This latter group of patients constitutes a major source of concern, bearing in mind that complementary medicine practitioners in SSA are likely to be unequipped to spot cancer or to make a specialist referral when necessary.

However, there is a need for further research to fully understand patients' pathways to cancer diagnosis in SSA. For instance, our review found that the majority of patients initially presented in primary care, but we are uncertain on the exact roles this played in their journey to diagnosis. As such, a comprehensive research programme to examine the role of primary care and alternative care in

cancer diagnosis is recommended as this may contribute to the development of possible diagnostic guidelines.

**Contributors** TM was involved in all aspects. WH participated in the study design, data interpretation and preparation and revision of the manuscript. SWDM participated in the assessment of studies quality and revision of the manuscript. All authors read and approved the final manuscript.

**Funding** TM received funding from Cancer Research UK (CRUK) Population Research Committee (C56361/A26124). SWDM is supported by the Can Test Collaborative, which is funded by CRUK (C8640/A23385). WH is codirector of CanTest.

**Competing interests** None declared.

**Patient consent for publication** Not required.

**Provenance and peer review** Not commissioned; externally peer reviewed.

**Data availability statement** Data sharing is not applicable as no datasets are generated and/or analysed for this study. No data are available. No additional data available.

**ORCID iDs**
Tanimola Martins http://orcid.org/0000-0001-5226-4073
Samuel William David Merriel http://orcid.org/0000-0003-2919-9087
William Hamilton http://orcid.org/0000-0003-1611-1373

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
