## [Reviewer comments · BMJ Open]

ARTICLE DETAILS

TITLE (PROVISIONAL)	Routes to diagnosis of symptomatic cancer in Sub-Saharan Africa: a systematic review
AUTHORS	Martins, Tanimola; Merriel, Samuel; Hamilton, Willie

VERSION 1 – REVIEW

REVIEWER	Andre Ilbawi World Health Organization Switzerland
REVIEW RETURNED	14-Apr-2020

GENERAL COMMENTS	The authors have performed a high-quality systematic narrative review on diagnostic pathways for cancer in sub-Saharan Africa. This valuable manuscript is a valuable addition to the limited understanding of early diagnosis in the Region and less developed countries. Four major considerations can significantly improve the quality of the review: Scope of review: the manuscript strives to present diagnostic pathways; however, the majority of the published studies and the focus of the discussion are on first contact point with the health system rather than diagnostic pathways. Articulating the distinction can improve the clarity of the review and support the conclusions. A discussion of the difference between the two concepts as well as defining diagnosis as clinical or pathologic can provide greater clarity to what the study is summarizing and its relevance in public health planning. Inclusion of studies that report on screening programme: several key factors must be considered when presenting screening in this article: 1) consideration can be made to exclude such studies. The fundamental principle of screening programmes is that they detect asymptomatic cases, which is different than the title and core concepts of this review. 2) Cancer screening should not be presented a priority intervention or cause for late diagnosis in SSA or other less developed setting. There are extremely limited data (outside trial setting) in which screening programmes have been successfully implemented. As such, statements attributing late stage diagnosis should not be assumed to be due to limited screening availability. 3) Cancer screening is the diagnostic pathway for <5% of cancers in HIC and should not be the focus of the discussion on pathways in LMIC where such programmes are absent or suffer low participation. 4) Cancer screening biases the entry point and diagnostic pathway for cancer patients. 5) Are the patients being seen in HIV clinics being screened for AIDS-related cancers? If so, this should also be stated more explicitly. Setting the stage: it would be worthwhile to present relevant data on
--

	diagnostic pathways for communicable diseases and/or data for pathways outside SSA. Such information can help justify this study and explain why such pathways are relevant. Research priorities: given the deficits in available data, it would be worthwhile for the authors to share their perspective on what are research priorities. Policy-makers in SSA are unlikely to endorse the proposed radical interventions without clarity on the impact, feasibility and cost of such programmes. Defining a research agenda can help gather necessary data to support policy formulation. Three additional minor consideration that can be made are: A more detailed discussion of / and precision in alternative medicine. Can the authors gather any data as to whether these “alternative medicines” or “providers” are in the formal health sector for a particular country? Why is it prevalent in some countries; could it be related to service availability? Feedback on search strategy: do the authors feel confident that studies reporting on patients who directly present to tertiary care have been included in this search strategy? Difference between urban and rural: did the authors identify any differences in diagnostic pathways between those living in urban vs rural settings? A comment on this would be important for the discussion and recommendations. Including these edits will likely strengthen the discussion and conclusion of this manuscript and will increase the impact of its findings. Stronger recommendations are needed and can be made.
--	---

REVIEWER	Sophie Pilleron Dept of Public Health, Univ of Otago, Wellington, New Zealand
REVIEW RETURNED	24-May-2020

GENERAL COMMENTS	General comments I thank authors to contribute to a better knowledge of cancer in Sub-Saharan Africa. A good description of the different routes taken by patients to reach cancer diagnosis is of importance to develop appropriate interventions to shorten the delay to be diagnosed. I acknowledge the huge amount of work behind this publication. However, the paper would deserve a bit more work to be improved. Below my recommendations. Abstract: Background: - « While effective public health measures used in developed counties can help minimise cancer risks, targeted and more radical approaches will be required to reduce cancer mortality in the region. »: It is not clear how this sentence introduces the objective of the paper. I would suggest to be more specific. - What is the justification of included papers reporting results from pilot study of assessment of an intervention? This does not seem to reflect the current practice? - Conclusion: overall, the conclusion does not come directly from results. I would suggest to focus it on the main objective of the study. Strengths and limitations of the study (and in the dedicated part in the discussion): - Because there were no quality check at each step of the selection of articles, authors cannot write that their methods followed best practice. However, they should highlight this point as a limitation.
---

	Introduction: The introduction would deserve to be shortened a bit and to be more focused on the route of diagnosis. Some parts of the introduction would be better placed in the discussion. Methods  - I would suggest the authors avoiding bullet points. They should consider writing full sentences instead. - The authors mentioned to identify predominant themes, but, if I am not wrong, I have not seen any mention of these themes in the results part. Results:  - Assessment of study quality: Authors mentioned that recruitment at hospitals may include a selection bias. This is right, however, it is difficult to recruit patients outside the hospital as cancer is a relatively rare disease. Another way, to recruit patients would be through cancer registries, but all sub-Saharan countries do not have a population-based cancer registry, and this would pose ethical issues. I do not think there is another mean to recruit patients with cancer. This should be acknowledged. - Also, the authors highlighted that most studies were descriptive. I am not sure how they rated the studies' quality against this criteria, but I would expect studies about the route to diagnosis to be descriptive. This is not a bad point. I would not consider these 2 last points « flaws ». - Authors made the distinction between diagnosis by physician and diagnosis made at the hospital. Do many physicians practice outside hospitals in the countries investigated? Discussion:  - I do not think that Egypt, Morocco, Algeria, Tunisia, or Libya are part of sub-Saharan Africa. I do not see the need to mention that you have excluded them as the review focused on sub-Saharan Africa. Their exclusion may be mentioned in the method part, though. - However, authors should have commented on the fact of excluding papers in French. Many researchers in French-speaking countries do not publish in English. Their exclusion is a limitation to discuss. - Authors discussed screening as a route to diagnosis. First, the aim of cervical cancer screening is not to detect cancerous lesions but to detect precancerous lesions. Screening may, indeed, detect some cancers. From the review, it is not clear if cervical cancer screening was already implemented in the country investigated or if the study was assessing the feasibility of such a screening method. This should be mentioned because I am not sure that if there was no screening in place, the study should be included. In addition, the authors stated they were interested in the diagnosis of cancer among symptomatic patients. Screening detects cancer in asymptomatic patients. - Authors discussed the importance of alternative medicine in SSA. I really liked this part. I would, however, suggest authors taking out the sentence Lines 33-38 starting by « The use of alternative medicine is not limited to SSA.... » as it does not add to their argument.
--	--

	- The message in the last paragraph is not clear. Authors only reported data available in the selected studies. They did not estimate anything, and they were not expected to do so. I would suggest to reformulate or to take out. Finally, the manuscript should be reviewed thoroughly for grammar and spelling mistakes. Overall, I think this is a good study, and I encourage authors to consider my recommendations.
--	--

VERSION 1 – AUTHOR RESPONSE

Reviewer: 1

Reviewer Name: Andre Ilbawi

Institution and Country:

World Health Organization

Switzerland

Please state any competing interests or state 'None declared': None declared

Please leave your comments for the authors below The authors have performed a high-quality systematic narrative review on diagnostic pathways for cancer in sub-Saharan Africa. This valuable manuscript is a valuable addition to the limited understanding of early diagnosis in the Region and less developed countries.

Four major considerations can significantly improve the quality of the review:

Scope of review: the manuscript strives to present diagnostic pathways; however, the majority of the published studies and the focus of the discussion are on first contact point with the health system rather than diagnostic pathways. Articulating the distinction can improve the clarity of the review and support the conclusions. A discussion of the difference between the two concepts as well as defining diagnosis as clinical or pathologic can provide greater clarity to what the study is summarizing and its relevance in public health planning.

Inclusion of studies that report on screening programme: several key factors must be considered when presenting screening in this article: 1) consideration can be made to exclude such studies. The fundamental principle of screening programmes is that they detect asymptomatic cases, which is different than the title and core concepts of this review. 2) Cancer screening should not be presented a priority intervention or cause for late diagnosis in SSA or other less developed setting. There are extremely limited data (outside trial setting) in which screening programmes have been successfully implemented. As such, statements attributing late stage diagnosis should not be assumed to be due to limited screening availability. 3) Cancer screening is the diagnostic pathway for <5% of cancers in HIC and should not be the focus of the discussion on pathways in LMIC where such programmes are absent or suffer low participation. 4) Cancer screening biases the entry point and diagnostic pathway for cancer patients.

Response: The above comments are very helpful, thank you. We have now excluded studies on cancer screening and updated the PRISMA flow chart and our results section as appropriate.

5) Are the patients being seen in HIV clinics being screened for AIDS-related cancers? If so, this should also be stated more explicitly.

Response: We have added the following statement to the results on Kaposi's sarcoma "In both studies, providers at the HIV clinics detected Kaposi's sarcomas during routine examination for opportunistic infections."

Setting the stage: it would be worthwhile to present relevant data on diagnostic pathways for communicable diseases and/or data for pathways outside SSA. Such information can help justify this study and explain why such pathways are relevant.

Response: Agree, we have edited the background and discussion sections in line with this comment.

Research priorities: given the deficits in available data, it would be worthwhile for the authors to share their perspective on what are research priorities. Policy-makers in SSA are unlikely to endorse the proposed radical interventions without clarity on the impact, feasibility and cost of such programmes. Defining a research agenda can help gather necessary data to support policy formulation.

Response: Agree, we have now included suggestions for possible research in the conclusion sections.

Three additional minor consideration that can be made are:

A more detailed discussion of / and precision in alternative medicine. Can the authors gather any data as to whether these “alternative medicines” or “providers” are in the formal health sector for a particular country? Why is it prevalent in some countries; could it be related to service availability?

Response: Again the reviewer raised an important point here. However, alternative medicine is complex, and going into detail regarding the degree of integration in SSA may deflect the review away from its original aim, thereby make the paper look like an advocate for alternative medicine. We have added the following statement to the discussion instead: “The use of alternative medicine is widespread in SSA, though not as fully developed compared to the practice in Asia and North America.”

Feedback on search strategy: do the authors feel confident that studies reporting on patients who directly present to tertiary care have been included in this search strategy?

Response: We are not fully confident on this point, and so added the following statement to the discussion section for clarity. “Also, we may have omitted some studies reporting cancer emergencies or direct symptomatic presentations to tertiary healthcare.”

Difference between urban and rural: did the authors identify any differences in diagnostic pathways between those living in urban vs rural settings? A comment on this would be important for the discussion and recommendations.

Response: The reviewer raised another interesting point here. However, our final selection did not specify where patients were resident at the time of diagnosis. So, it is difficult for us to assume that the route to diagnosis differs based on urban/rural dwelling. We have now included a statement in the discussion reflecting the additional difficulty rural dwelling poses to cancer diagnoses.

Reviewer: 2

Reviewer Name: Sophie Pilleron

Institution and Country: Dept of Public Health, Univ of Otago, Wellington, New Zealand Please state any competing interests or state ‘None declared’: None declared

Please leave your comments for the authors below Review BMJ

General comments

I thank authors to contribute to a better knowledge of cancer in Sub-Saharan Africa. A good description of the different routes taken by patients to reach cancer diagnosis is of importance to develop appropriate interventions to shorten the delay to be diagnosed. I acknowledge the huge amount of work behind this publication. However, the paper would deserve a bit more work to be improved. Below my recommendations.

Abstract:

Background:

- « While effective public health measures used in developed counties can help minimise cancer risks, targeted and more radical approaches will be required to reduce cancer mortality in the region. »: It is not clear how this sentence introduces the objective of the paper. I would suggest to be more specific.

Response: Agree, we have edited this section with the following statements: “Most cancers in Sub-Saharan Africa (SSA) are diagnosed at advanced stages, with limited treatment options and poorer outcomes. Part of this may be linked to various events occurring in patients’ journey to diagnosis. Using the Model of Pathways to Treatment we examined the evidence regarding the routes to diagnosis of cancer in SSA”.

- What is the justification of included papers reporting results from pilot study of assessment of an intervention? This does not seem to reflect the current practice?

Response: *Agree. Both pilot studies have now been excluded, thank you.*

- Conclusion: overall, the conclusion does not come directly from results. I would suggest to focus it on the main objective of the study.

Response: *Agree, we have now revised this section with the following: "In their journey to diagnosis, patients in SSA initially consult in primary care or use alternative medicine. Government and health departments in SSA must find radical solutions to the rising burden of cancer in the region. Investment in sustained cancer awareness programme, research, training and development of primary care and alternative medicine providers to spot and refer suspected cases early, may help improve cancer outcomes in the region".*

Strengths and limitations of the study (and in the dedicated part in the discussion):

- Because there were no quality check at each step of the selection of articles, authors cannot write that their methods followed best practice. However, they should highlight this point as a limitation.

Response: *We have now performed quality assessments of studies using the Newcastle-Ottawa Quality Assessment Scale (NOS) for cohort, NOS adapted for cross-sectional studies, and the Joanna Briggs Institute (JBI) Critical Appraisal Checklist for Qualitative Research.*

Introduction:

The introduction would deserve to be shortened a bit and to be more focused on the route of diagnosis. Some parts of the introduction would be better placed in the discussion.

Response: *We have revised this section considerably, thanks*

Methods

- I would suggest the authors avoiding bullet points. They should consider writing full sentences instead.

Response: *Agree, inclusion and exclusion criteria now writing in full sentences.*

Results:

- Assessment of study quality: Authors mentioned that recruitment at hospitals may include a selection bias. This is right, however, it is difficult to recruit patients outside the hospital as cancer is a relatively rare disease. Another way, to recruit patients would be through cancer registries, but all sub-Saharan countries do not have a population-based cancer registry, and this would pose ethical issues. I do not think there is another mean to recruit patients with cancer. This should be acknowledged.

Response: *Agree, we have added the following statement: "This is not surprising, with weak primary care services and the absence of established cancer registries in SSA, thus limiting the quality and quantity of data available for research."*

- Also, the authors highlighted that most studies were descriptive. I am not sure how they rated the studies' quality against this criteria, but I would expect studies about the route to diagnosis to be descriptive. This is not a bad point. I would not consider these 2 last points « flaws ».

Response: *Agree, thank you.*

- Authors made the distinction between diagnosis by physician and diagnosis made at the hospital. Do many physicians practice outside hospitals in the countries investigated?

Response: *Indeed, a considerable proportion of physicians work in private practices.*

Discussion:

- I do not think that Egypt, Morocco, Algeria, Tunisia, or Libya are part of sub-Saharan Africa. I do not see the need to mention that you have excluded them as the review focused on sub-Saharan Africa. Their exclusion may be mentioned in the method part, though.

- However, authors should have commented on the fact of excluding papers in French. Many researchers in French-speaking countries do not publish in English. Their exclusion is a limitation to discuss.

Response: *Agree, we have added the following statement to the discussion: “Additionally, our search strategies omitted non-English studies, thereby excluding any studies published in French from the small number of Francophone countries in SSA. The decision to omit these countries in our search strategy may have reduced the number of selected studies slightly, but we have no reason to believe that such omission had any impact on our findings”.*

- Authors discussed screening as a route to diagnosis. First, the aim of cervical cancer screening is not to detect cancerous lesions but to detect precancerous lesions. Screening may, indeed, detect some cancers. From the review, it is not clear if cervical cancer screening was already implemented in the country investigated or if the study was assessing the feasibility of such a screening method. This should be mentioned because I am not sure that if there was no screening in place, the study should be included. In addition, the authors stated they were interested in the diagnosis of cancer among symptomatic patients. Screening detects cancer in asymptomatic patients.

Response: *We have now excluded all screening studies*

- Authors discussed the importance of alternative medicine in SSA. I really liked this part. I would, however, suggest authors taking out the sentence Lines 33-38 starting by « The use of alternative medicine is not limited to SSA... » as it does not add to their argument.

Response: *We have now edited the sentence as follows “The use of alternative medicine is widespread in SSA, though not as fully developed compared to the practice in Asia and North America.”*

- The message in the last paragraph is not clear. Authors only reported data available in the selected studies. They did not estimate anything, and they were not expected to do so. I would suggest to reformulate or to take out.

Response: *We have now revised the paragraph, thank you.*

Finally, the manuscript should be reviewed thoroughly for grammar and spelling mistakes.

Response: *Agree*

Overall, I think this is a good study, and I encourage authors to consider my recommendations.

Response: *The recommendations have helped improve the manuscript considerably, thank you.*

VERSION 2 – REVIEW

REVIEWER	Sophie Pilleron NDPH, Univ of Oxford
REVIEW RETURNED	19-Aug-2020

GENERAL COMMENTS	Routes to diagnosis of symptomatic cancer in Sub-Saharan Africa: a systematic review I thank the editors for the opportunity to review this new version of the manuscript. I thank the authors for taking into account the comments of the previous review. The paper is much improved. However, I still have minor comments. Abstract: In the conclusion, the following sentence « Government and health departments in Sub-Saharan Africa must find radical solutions to the rising burden of cancer in the region. » does not come directly from the results of the literature review. Please, focus it on the main objective of the study. Introduction: « These figures are considerably higher than the estimates for
---

2012,5 »: 2018 estimates cannot be compared with 2012 estimates as methods may have been changed, more registries may have been included, etc. Please, consider take out this sentence.

« some of which may be addressed by implementing effective public health interventions. »: even if this statement is true, this is summarizing very quickly the reality. Many other considerations should be taken into account.

The introduction may be further shortened (take out the part about incidence and mortality) to focus on the topic of the paper.

Methods

I acknowledge that the authors performed a second review of all papers against quality criteria, that improved the quality of the study considerably. However, best practice requests that at least two people do the screening of all references at each step of the selection. I understand that it was not the case of this study.

I would advise authors to add somewhere they included all studies regardless of the study design (i.e. quantitative and qualitative studies). Maybe this should be in the « eligibility criteria » part.

Results

Again, the quality criteria assessed are not suitable for descriptive studies. Authors are talking about « flaws » while what is appearing as limitations are actually inherent in the descriptive study design. This is not bad quality studies. I would, therefore, advise revising this part.

Discussion

Line 2: authors examined evidence about the route to diagnosis and not "cancer diagnosis", a term that is not specific enough and covers more than the journey to diagnosis.

Strength and limitation: Again, your methodology does not precisely follow best practice (I acknowledge that in practice, best practice is difficult to follow).

I would take out the following sentences as they do not bring much:
« We omitted the British and French Overseas Territories and few countries (Egypt, Morocco, Algeria, Tunisia, and Libya), which are arbitrarily usually classed as part of the Arab world. Health services in some of these countries are similar to those of the developed world, providing universal care through social or government contributions.54 55

« Healthcare services in many SSA countries are not universally accessible. They are pluralistic with a range of public and private providers who barely communicate with each other. » should be moved elsewhere.

« ...thereby excluding any studies published in French from a small number of Francophone countries in SSA » is not useful, as non-English studies include French studies and possible studies in other languages as well (i.e. Portuguese).

	The fact that half of the studies were about breast cancer is a result, not a limitation. I would remove the beginning of the sentence. If authors systematically searched the studies about the route to diagnosis, it is not clear why the authors would have omitted « some studies reporting cancer emergencies or direct symptomatic presentations to tertiary healthcare ». Authors are wrong when they write that there are no cancer registries in SSA. Please, visit https://afcrn.org. Some of these registries are of good quality and are included in the CI5 collection. Authors did not understand my comment in my previous review. They highlighted recruitment in hospitals as a limitation. Once again, there are not a lot of options to recruit patients with cancer. I understand that affordability may limit access to hospitals; I have more difficulties in understanding why comorbidity would prevent a cancer diagnosis. Other factors have a bigger weight in SSA such as financial and physical accessibility. « ...though not as fully developed compared to the practice in Asia and North America » does not add to the discussion. To me, the last paragraph of the discussion would be better placed in the limitation part. However, as said in my previous review, this study aimed at examining literature regarding the route to diagnosis in SSA, not to estimate any proportion. Authors mentioned fairly limitations of studies, notably the lack of generalizability earlier in the discussion. I would take this part out. The conclusion is much too long. It should be 2-3 sentences, not more. As is, it would be better placed in the discussion. However, the discussion is already very long. I would advise shortening the discussion a bit. I have already suggested some texts that could be easily taken out. The conclusion should be more in line with the results of the review, and opening on further research (as authors did but shorter).
--	--

VERSION 2 – AUTHOR RESPONSE

Reviewer Name

Sophie Pilleron

Institution and Country

NDPH, Univ of Oxford

Please state any competing interests or state 'None declared':
None

Routes to diagnosis of symptomatic cancer in Sub-Saharan Africa: a systematic review

I thank the editors for the opportunity to review this new version of the manuscript.
I thank the authors for taking into account the comments of the previous review. The paper is much

improved. However, I still have minor comments.

Abstract:

In the conclusion, the following sentence « Government and health departments in Sub-Saharan Africa must find radical solutions to the rising burden of cancer in the region. » does not come directly from the results of the literature review. Please, focus it on the main objective of the study.

Response: We have now revised the conclusion section.

Introduction:

« These figures are considerably higher than the estimates for 2012,5 »: 2018 estimates cannot be compared with 2012 estimates as methods may have been changed, more registries may have been included, etc. Please, consider take out this sentence.

Response: We have now removed the above sentence.

« some of which may be addressed by implementing effective public health interventions. »: even if this statement is true, this is summarizing very quickly the reality. Many other considerations should be taken into account.

The introduction may be further shortened (take out the part about incidence and mortality) to focus on the topic of the paper.

Response: We have now revised the introduction in line with the above suggestion, thank you.

Methods

I acknowledge that the authors performed a second review of all papers against quality criteria that improved the quality of the study considerably. However, best practice requests that at least two people do the screening of all references at each step of the selection. I understand that it was not the case of this study.

I would advise authors to add somewhere they included all studies regardless of the study design (i.e. quantitative and qualitative studies). Maybe this should be in the « eligibility criteria » part.

Response: We have replaced best with “good practice” and added the following sentence to the methods. “All study designs (qualitative and quantitative) were eligible for inclusion”.

Results

Again, the quality criteria assessed are not suitable for descriptive studies. Authors are talking about « flaws » while what is appearing as limitations are actually inherent in the descriptive study design. This is not bad quality studies. I would, therefore, advise revising this part.

Response: We have now revised this aspect and the term “flaws” replaced with limitations/limits as appropriate.

Discussion

Line 2: authors examined evidence about the route to diagnosis and not "cancer diagnosis", a term that is not specific enough and covers more than the journey to diagnosis.

Strength and limitation: Again, your methodology does not precisely follow best practice (I acknowledge that in practice, best practice is difficult to follow).

Response: We have replaced “best” with “good practice” in line with the reviewers comment here.

I would take out the following sentences as they do not bring much: « We omitted the British and French Overseas Territories and few countries (Egypt, Morocco, Algeria, Tunisia, and Libya), which are arbitrarily usually classed as part of the Arab world. Health services in some of these countries

are similar to those of the developed world, providing universal care through social or government contributions.^{54 55}

« Healthcare services in many SSA countries are not universally accessible. They are pluralistic with a range of public and private providers who barely communicate with each other. » should be moved elsewhere.

Response: We have now removed the above statements.

« ...thereby excluding any studies published in French from a small number of Francophone countries in SSA » is not useful, as non-English studies include French studies and possible studies in other languages as well (i.e. Portuguese).

Response: We have now revised this statement with the following: "We omitted non-English studies as these may include studies published in French, Portuguese, and other African languages."

The fact that half of the studies were about breast cancer is a result, not a limitation. I would remove the beginning of the sentence.

Response: We have now revised this statement with the following: "About half of the studies focused on breast cancer, reducing the scope of the review".

If authors systematically searched the studies about the route to diagnosis, it is not clear why the authors would have omitted « some studies reporting cancer emergencies or direct symptomatic presentations to tertiary healthcare ».

Response: We wanted to acknowledge the limitations in our search strategy, and in the doing so we became our own worst critic. We have now removed the statement relating to the above sentence. Thanks

Authors are wrong when they write that there are no cancer registries in SSA. Please, visit <https://afcrn.org>. Some of these registries are of good quality and are included in the CI5 collection.

Response: We have now revised the statement suggesting the absence of established cancer registry in SSA as follows. "This is not surprising, given the weak primary care and limited cancer registries in SSA, thus limiting the quality and quantity of data available for research."

Authors did not understand my comment in my previous review. They highlighted recruitment in hospitals as a limitation. Once again, there are not a loof options to recruit patients with cancer.

Response: While this may be less apparent in SSA, due to the health systems organisation, the journey to cancer diagnosis often starts with the patient recognising and presenting potential symptoms in primary care settings. Therefore, recruiting participants in this setting (as against hospital settings) provides a unique opportunity to learn more about the pathways to diagnosis, and areas in the pathways where effective interventions can be targeted.

I understand that affordability may limit access to hospitals; I have more difficulties in understanding why comorbidity would prevent a cancer diagnosis. Other factors have a bigger weight in SSA such as financial and physical accessibility.

Response: Misdiagnosis is not unusual in clinical practice, more so where the number of experts and diagnostic facilities are limited. Most cancers present with non-specific symptoms - for instance cough, weight loss and abdominal pain - which are associated with benign diseases. A patient with known COPD, for example, may be misdiagnosed or (never be investigated for possible cancer) if they present with cough or weight loss. Such patients in SSA may die of stroke or other vascular disease (which are common in the region), and their cancer missed completely. Thus comorbidity plays a key role in cancer diagnosis. There is a growing body of evidence on this in the developed world: we've not added references from that arena, as they are peripheral to the main thrust of the review (we can do so if the editor prefers us to).

« ...though not as fully developed compared to the practice in Asia and North America » does not add to the discussion.

Response: We have now revised the statement as follows: “The use of alternative medicine is widespread in SSA, although evidence suggests that the practitioners can to misdiagnose cancer, resulting in advanced-stage diagnosis and reduced chances of survival.”

To me, the last paragraph of the discussion would be better placed in the limitation part. However, as said in my previous review, this study aimed at examining literature regarding the route to diagnosis in SSA, not to estimate any proportion. Authors mentioned fairly limitations of studies, notably the lack of generalizability earlier in the discussion. I would take this part out.

The conclusion is much too long. It should be 2-3 sentences, not more. As is, it would be better placed in the discussion. However, the discussion is already very long. I would advise shortening the discussion a bit. I have already suggested some texts that could be easily taken out.

The conclusion should be more in line with the results of the review, and opening on further research (as authors did but shorter).

We have now revised the conclusion section in line with this above suggestions, thank you.

VERSION 3 – REVIEW

REVIEWER	Sophie Pilleron NDPH, University of Oxford
REVIEW RETURNED	12-Oct-2020
GENERAL COMMENTS	No further comments. Thank you for the opportunity to review it